# Fast Anomaly Detection for Vision-Based Industrial Inspection Using Cascades of Null Subspace PCA Detectors

**DOI:** 10.3390/s25154853

**Published:** 2025-08-07

**Authors:** Muhammad Bilal, Muhammad Shehzad Hanif

**Affiliations:** 1Department of Electrical and Computer Engineering, College of Engineering, King Abdulaziz University, Jeddah 21589, Saudi Arabia; mshanif@kau.edu.sa; 2Center of Excellence in Intelligent Engineering Systems (CEIES), King Abdulaziz University, Jeddah 21589, Saudi Arabia

**Keywords:** computer vision, anomaly detection, industrial inspection, Principal Component Analysis, null subspace, cascaded detectors

## Abstract

Anomaly detection in industrial imaging is critical for ensuring quality and reliability in automated manufacturing processes. While recently several methods have been reported in the literature that have demonstrated impressive detection performance on standard benchmarks, they necessarily rely on computationally intensive CNN architectures and post-processing techniques, necessitating access to high-end GPU hardware and limiting practical deployment in resource-constrained settings. In this study, we introduce a novel anomaly detection framework that leverages feature maps from a lightweight convolutional neural network (CNN) backbone, MobileNetV2, and cascaded detection to achieve notable accuracy as well as computational efficiency. The core of our method consists of two main components. First is a PCA-based anomaly detection module that specifically exploits near-zero variance features. Contrary to traditional PCA methods, which tend to focus on the high-variance directions that encapsulate the dominant patterns in normal data, our approach demonstrates that the lower variance directions (which are typically ignored) form an approximate null space where normal samples project near zero. However, the anomalous samples, due to their inherent deviations from the norm, lead to projections with significantly higher magnitudes in this space. This insight not only enhances sensitivity to true anomalies but also reduces computational complexity by eliminating the need for operations such as matrix inversion or the calculation of Mahalanobis distances for correlated features otherwise needed when normal behavior is modeled as Gaussian distribution. Second, our framework consists of a cascaded multi-stage decision process. Instead of combining features across layers, we treat the local features extracted from each layer as independent stages within a cascade. This cascading mechanism not only simplifies the computations at each stage by quickly eliminating clear cases but also progressively refines the anomaly decision, leading to enhanced overall accuracy. Experimental evaluations on MVTec and VisA benchmark datasets demonstrate that our proposed approach achieves superior anomaly detection performance (99.4% and 91.7% AUROC respectively) while maintaining a lower computational overhead compared to other methods. This framework provides a compelling solution for practical anomaly detection challenges in diverse application domains where competitive accuracy is needed at the expense of minimal hardware resources.

## 1. Introduction

Industrial anomaly detection is crucial for ensuring quality and efficiency in modern manufacturing processes, where even minor defects can lead to significant operational and economic losses. Anomalies encountered in industrial settings are diverse and unpredictable, representing instances that contain patterns deviating from those seen in normal instances. Localized and fine-grained structural anomalies are physical defects or inconsistencies in the product’s material or surface e.g., (i) visible contaminations or stains in soaps, capsules and pills etc. (ii) cracks in items like bottles or wood (iii) scratches, holes and twists etc. as seen on toothbrushes or grids (iv) missing components or broken parts such as cracked glass, broken leads on a transistor, or broken teeth on a zipper. Similarly, various material inconsistencies like bent wire or cable swaps and other surface irregularities including print errors, glue, or oil can also be considered anomalies in industrial production lines. More subtle examples also include dislocated or flipped components, for instance, on a metal nut. However, even speckle noise in the background can sometimes be misidentified as an anomaly. In contrast, logical anomalies represent violations of learned logical constraints concerning the position, size, or arrangement of otherwise normal objects within an image, posing a more significant detection challenge. Examples of these include: a wrong ordering of objects; an incorrect combination of otherwise normal objects; missing objects; misplaced objects; surplus objects; or a violation of geometrical constraints, such as a screw being two millimeters too long. Thus, industrial anomalies being subtle and often being in close proximity to normal regions pose a significant detection problem.

In this regard, recent advances in machine vision to detect anomalous patterns have seen the proliferation of deep learning techniques, with many state-of-the-art methods employing complex architectures like ResNet18, ResNet50, or ResNet101 [1] to extract robust features from high-resolution images. The use of these CNNs to extract features has been paramount due to their effectiveness in capturing generalized, robust and multi-scale representations of images since these are pre-trained on large datasets like ImageNet. Due to the diverse nature of ImageNet, these features capture a wide range of visual information, from low-level textures to high-level semantic information. This capability is essential for distinguishing subtle imperfections from normal variations since they provide a hierarchy of feature maps from their intermediate layers, which represent information at different levels of abstraction and resolution [2]. This multi-scale nature is critical for robust anomaly detection, as it allows for the identification of various defect types, from minute, fine-grained flaws (e.g., subtle scratches captured by high-resolution features) to larger, structural deviations (captured by high-level semantic features). A crucial aspect in utilizing pre-trained CNNs is that this approach avoids the need for extensive training of the backbone network on specific industrial datasets as required by some recent methods [3,4,5], reducing training overhead and simplifying the anomaly detection pipeline. Moreover, the industrial benchmarking datasets do not contain extensive examples to train a large CNN and hence may lead to overfitting. Furthermore, the well-defined receptive field of CNNs, ensures that anomalies in one part of an image do not trigger anomalous feature vectors in distant parts, which is crucial for accurate localization.

Central to our method is the utilization of near-zero variance features that define the null subspace of normal data [6], where typical images project near zero while anomalies yield significantly higher responses. A similar approach has been earlier applied for anomaly detection in hyperspectral images [7]. Furthermore, we introduce a cascaded detection strategy in which the anomaly scores derived from each CNN layer are evaluated independently at successive stages. At each stage, dual thresholds rapidly classify definitive normals and anomalies, while only ambiguous cases are passed to the next level, thereby streamlining computations and progressively refining the detection process. Through this novel combination of near-zero variance PCA-based analysis and cascaded decision-making, our approach achieves the dual goal of high accuracy and reduced computational complexity, yielding a for more effective and scalable solutions in vision-based industrial inspection.

The main contributions of this work can be summarized as follows:We introduce a novel anomaly detection framework for industrial inspection that leverages the compact yet robust feature representations of a pretrained MobileNetV2 (known for small memory and computing footprint), eliminating the need for large, complex backbones. Source codes are openly available: https://github.com/4mbilal/Anomaly_Detection (accessed on 22 June 2025).We propose a PCA-based technique that specifically exploits near-zero variance features, effectively identifying the null subspace where normal samples project near zero while anomalies stand out, thus enabling efficient and accurate detection.We develop a cascaded multi-stage detection strategy, where dual thresholds at each stage rapidly classify clearly normal or anomalous samples, and only ambiguous cases are forwarded for further analysis. This progressive filtering reduces computational load and refines decision boundaries.We address practical pre-processing challenges by incorporating tailored image padding and augmentation techniques that mitigate distortion and enhance dataset diversity, especially for classes with limited sample sizes.Experimental results on industrial inspection datasets demonstrate that our combined approach not only achieves high accuracy but also significantly reduces computational complexity (*20.1* fps on a low-end GPU), making it highly suitable for real-time applications.

## 2. Literature Review

Anomaly detection and localization in industrial environments presents a crucial challenge focused on identifying and precisely pinpointing abnormal regions using only data from normal, defect-free samples. This task is essential for quality control in manufacturing, as traditional manual inspection methods are both time-consuming and susceptible to human error, highlighting the need for efficient automated systems. A significant hurdle in this field is the inherent scarcity of actual defective samples and the high cost associated with obtaining detailed pixel-level annotations for them, which largely precludes the use of fully supervised learning approaches. Consequently, models are often trained in a “one-class learning” setting, where the system must learn to classify anything deviating from the normal data distribution as anomalous, despite anomalies potentially exhibiting a wide range of “unexpected patterns,” from subtle scratches to major structural flaws.

To address these complexities, researchers have predominantly pursued three main methodological directions as discussed in the following subsections.

### 2.1. Reconstruction-Based Methods

Reconstruction-based methods are a prominent category in anomaly detection and localization, operating on the fundamental assumption that a model trained exclusively on normal, non-defective samples will accurately reconstruct only these normal regions, while failing to reconstruct abnormal or anomalous inputs. The core idea is to identify anomalies by analyzing the differences between the original image and its reconstructed version. However, a limitation arises if the model becomes too adept at reconstructing even anomalous inputs, leading to missed detections.

Common neural network architectures employed in this approach include AutoEncoders (AEs) [8,9], Variational AutoEncoders (VAEs) [10], and Generative Adversarial Networks (GANs) [11,12,13]. Cheng et al. have applied AE-based anomaly detection technique to hyperspectral images as well [14]. Some methods may also frame anomaly detection as an inpainting problem, where parts of images are randomly masked, and a neural network is used to predict the erased information. The anomaly score is typically derived from the residual image, or the pixel-wise reconstruction error, such as the L2 error. Additional information from the latent space, intermediate activations, or a discriminator can also be used to enhance anomaly recognition. Structural similarity index (SSIM) loss functions are also widely used during training.

Despite being intuitive and interpretable, reconstruction-based methods face limitations. A significant challenge is that the trained models can sometimes “generalize” too well, leading to accurate reconstruction of anomalous images, which results in missed detections. This means the assumption that anomalous regions will not be properly reconstructed may not always hold. Additionally, the performance of these methods heavily relies on the quality of the reconstructed image and the effectiveness of the difference analysis method used.

### 2.2. Embedding-Based Methods

Embedding-based methods represent a significant direction in anomaly detection and localization, primarily by leveraging pre-trained networks to transform input images into a compact feature space. The core principle behind these approaches is to learn a representation where features of normal, defect-free samples cluster tightly, while anomalous features are distinctly separated or lie outside this learned distribution.

These methods commonly employ Convolutional Neural Networks (CNNs), often pre-trained on large datasets like ImageNet, to extract generalized features. The extracted features, typically from intermediate layers, are then used to model the characteristics of normality. Several techniques are utilized for this purpose:*Memory Bank Methods* like SPADE [2] and PatchCore [15] store representative normal features in a “memory bank” similar to K-Nearest Neighbor (KNN) in nature. PatchCore, for instance, focuses on locally aggregated, mid-level feature patches and employs greedy coreset subsampling to reduce redundancy in the memory bank, thereby minimizing storage memory and inference time, which is highly beneficial for industrial applications. Anomaly detection is then performed by comparing input features to these memorized normal features, often using distance metrics like $\ell_{2}$-distance.*Statistical Distribution Modeling*-based methods such as PaDiM [16], describe the normal class through a set of multivariate Gaussian distributions, where each patch position in the feature map is associated with its own distribution. This method is designed to have low time and space complexity at test time, independent of the training dataset size, addressing scalability concerns of KNN-based methods.*One-Class Classification (OCC)* methods explicitly define classification boundaries, such as hyperplanes or hyperspheres [17], to distinguish normal from anomalous data.*Distribution Transformation* methods such as Normalizing Flow (NF) aim to transform the distribution of normal samples into a standard Gaussian distribution, where anomalies are then identified by their low likelihood in this transformed space [18].*Knowledge Distillation*-based approach involves training a student network to mimic the outputs of a fixed pre-trained teacher network using only normal samples [19]. Anomalies are detected by observing discrepancies between the teacher’s and student’s outputs.

While embedding-based methods have achieved strong performance, they face several challenges. Many of these methods can suffer from high computational complexity or high memory consumption, which can hinder practical industrial deployment. For instance, K-NN based methods exhibit linear complexity, meaning their time and space requirements increase with the size of the training dataset, posing scalability issues. Another concern is the potential for “mismatch problems” or “domain bias” when directly applying ImageNet pre-trained features to industrial images, as their data distributions may differ significantly. Some later innovations, like SimpleNet [3] and SuperSimpleNet [4], address this by introducing a “feature adaptor” to transfer features to the target domain and generating anomalies in the feature space rather than directly on images. Additionally, normalizing flow methods can be memory intensive, and knowledge distillation methods may increase computational complexity during inference. Furthermore, some embedding similarity-based methods, particularly those focused solely on detection, may lack interpretability regarding the specific anomalous regions within an image.

### 2.3. Synthesis-Based Methods

These methods are a distinct category in anomaly detection and localization, operating on the principle of generating artificial anomalous samples from normal data to introduce discriminative information into the detection model. This strategy is particularly valuable because collecting sufficient real defect samples and obtaining costly pixel-level annotations for them is often impractical in industrial settings. By synthesizing anomalies, these methods aim to mitigate potential overfitting that could occur when models are trained only on normal samples.

These methods can broadly be categorized into two main approaches:*Image-level anomaly synthesis* explicitly simulates anomalies directly on the image itself. Techniques include cutting and pasting normal image regions at random positions, as seen in methods like CutPaste [20]. Other approaches involve seamlessly blending blocks from different images, such as NSA [21], or creating binary masks (e.g., using Perlin noise [4]) and filling them with external textures, such as DRAEM [22]. While this approach can provide detailed anomaly textures, it often suffers from a lack of diversity and realism in the synthesized anomalies. The synthetic appearances may not closely match real defects, and features derived from such synthetic data might deviate significantly from actual normal features, potentially resulting in a loosely bounded normal feature space that could inadvertently classify subtle defects as normal.*Feature-level anomaly synthesis* implicitly simulates anomalies within the feature space extracted by a neural network. This approach is generally more efficient due to the smaller size of feature maps compared to full images. SimpleNet, for instance, generates anomalous features by adding Gaussian noise to normal features, which have first been processed by a “feature adaptor” to reduce domain bias from pre-trained backbones. The network then trains a simple discriminator, often a multi-layer perceptron (MLP), to distinguish between these adapted normal features and the synthesized anomalous features. A key challenge for early feature-level methods was the lack of controllable and directional synthesis, particularly for anomalies that are very similar to normal regions. More advanced methods, such as Global and Local Anomaly co-Synthesis Strategy (GLASS) [23], address this by guiding Gaussian noise with gradient ascent and truncated projection to synthesize “near-in-distribution anomalies” in a controllable manner. This allows for the generation of both “weak anomalies” close to normal points and “strong anomalies” further away. SuperSimpleNet [4] also employs feature-space anomaly generation, using a binarized Perlin noise mask to define regions where Gaussian noise is applied to adapted features, leading to more realistic and spatially coherent synthetic anomalous regions.

A common component across many synthesis-based methods is a discriminator or segmentation head that learns to distinguish between normal and anomalous patterns based on the synthesized data. While SimpleNet uses a truncated L1 loss for this, GLASS incorporates Binary Cross-Entropy (BCE) and Focal loss along with Online Hard Example Mining (OHEM), to address class imbalance and focus on crucial samples like weak defects.

The main advantages of synthesis-based methods include their ability to provide anomaly discrimination information, thereby enhancing unsupervised detection, and their efficiency, especially when anomalies are generated at the feature level. SimpleNet is noted for its outstanding performance and fast inference speed. GLASS specifically excels in detecting weak defects, and SuperSimpleNet enhances training consistency and can unify unsupervised and supervised learning settings, improving performance and robustness. However, practical challenges remain, such as ensuring the realism and diversity of synthetic anomalies, accurately representing the vast and unpredictable range of real defects, and calibrating noise levels for feature-level synthesis. Some models, like SuperSimpleNet, can also be sensitive to the choice of backbone network and the scale of Gaussian noise, which may require adjustment per backbone.

### 2.4. Speed, Memory and Training Requirements

Several anomaly detection methods utilize specific CPU/GPU environments and CNN architectures to achieve their reported speeds and performance. For instance, PaDiM had reported its inference times only on a mainstream CPU, an Intel i7-4710HQ CPU @ 2.50 GHz. On this setup, PaDiM-R18-Rd100 and PaDiM-WR50-Rd550 took 0.23 and 0.95 s respectively per image detection. Since, PaDiM leverages pre-trained CNNs like ResNet (e.g., ResNet18), Wide-ResNet (e.g., Wide ResNet-50-2), or EfficientNet (e.g., EfficientNet-B5) for embedding extraction, this method does not require cumbersome deep neural network training.

PatchCore’s inference times were reported on a system with an Nvidia GeForce GTX 3080 ti GPU and an Intel(R) Xeon(R) CPU E5-2680 v3@2.5 GHZ. Mean inference times per image on the MVTec AD dataset ranged from 0.6 s for PatchCore-100% (without subsampling) to 0.17 s for PatchCore-1%. PatchCore-100% with approximate nearest neighbor search (IVFPQ) reduced the time to 0.2 s. In another benchmark [4], PatchCore was reported to have an inference time of 54.1 ± 1.36 ms and a throughput of 25.3 ± 0.64 fps on an NVIDIA Tesla V100S. PatchCore typically uses a WideResNet50 backbone, pre-trained on ImageNet. It aggregates patch-level features from intermediate or mid-level feature representations, specifically from blocks 2 and 3 of ResNet-like architectures. Similar to PaDiM, this method operates without requiring training on the dataset at hand and uses a memory bank of nominal patch-features. The memory bank itself is a collection of extracted features, not a set of learnable parameters in the neural network sense.

SimpleNet achieves an inference speed of 77 frames per second (fps) on an Nvidia GeForce GTX 3080 ti GPU, using an Intel(R) Xeon(R) CPU E5-2680 v3@2.5 GHZ. An optimized version of SimpleNet [4] was later reported to have an inference time of 22.4 ± 2.42 milliseconds (ms) and a throughput of 95.2 ± 3.1 images per second (fps) on an AMD Epyc 7272 CPU and an NVIDIA Tesla V100S GPU. For its architecture, SimpleNet typically employs a WideResNet50 backbone, pre-trained on ImageNet, to extract features from its 2nd and 3rd intermediate layers. While the feature extractor is a frozen pre-trained network, SimpleNet’s learnable parameters constitute a shallow Feature Adaptor (implemented as a single fully-connected layer without bias) and a simple Anomaly Discriminator (a 2-layer Multi-Layer Perceptron).

SuperSimpleNet, an improvement over SimpleNet, has been reported to achieve an even faster inference time of 9.3 ms and a throughput of 268 images per second. These measurements were conducted on an advanced system equipped with an NVIDIA Tesla V100S GPU and an AMD Epyc 7272 CPU. Similar to SimpleNet, SuperSimpleNet uses a ResNet-like convolutional neural network, specifically a WideResNet50 pre-trained on ImageNet, as its feature extractor. It extracts features from the 2nd and 3rd layers of this backbone. The model has a total of 34 million parameters, but only 9 million of these are trainable, as the rest belong to the frozen pre-trained backbone.

The GLASS framework was implemented and evaluated on an NVIDIA Tesla A800 GPU and an Intel(R) Xeon(R) Gold 6346 CPU @3.10 GHz. It demonstrated a high throughput of 1327 images per second across its different variants (GLASS-m, GLASS-h, GLASS-j). For its CNN backbone, GLASS uses WideResnet50 as the feature extractor, merging features from level 2 and level 3. The feature extractor is typically frozen, while the feature adaptor and the discriminator are trainable components.

In conclusion, while these methods achieve impressive AUROC scores (approaching 100% on datasets like MVTec AD [24] and VisA [25]), there remains a gap in optimizing their speed and memory consumption for deployment on resource-constrained devices with limited CPU/GPU and system memory. Despite high throughputs and optimizations like coreset subsampling, challenges related to computational complexity and memory usage persist, highlighting the need for further research in this area for practical industrial applications.

## 3. Materials and Methods

In this work, we propose a novel anomaly detection framework that leverages local features extracted from the intermediate layers of MobileNetV2—a lightweight and efficient CNN architecture. Unlike many previous approaches that employ more complex backbones such as ResNet18, ResNet50, or ResNet101, our method capitalizes on the compact representation and reduced computational as well as memory demands of MobileNetV2. Moreover, we leverage MobileNetV2 as a fixed feature extractor, using its rich, pretrained ImageNet features rather than retraining the network. This approach allows us to exploit well-established, robust representations that capture both low-level textures and high-level semantic information, streamlining our anomaly detection pipeline without additional training overhead.

Our proposed approach consists of two main components.

PCA-based anomaly detection module that exploits near-zero variance features: by focusing on the eigenvectors corresponding to small eigenvalues, we capture the null space where normal samples project near zero, while anomalous samples exhibit significant deviations.A cascaded multi-stage strategy: Instead of combining features from all layers in a single detector, we treat the features from each CNN layer independently in stages which employ dual thresholds to immediately classify clear cases and forward only ambiguous samples to the next stage.

These two strategies not only simplify the computations at each step, but also lead to overall improved detection accuracy by progressively refining the anomaly decision.

In the discussion that follows, the input *x* represents a local feature vector extracted from one of the various layers of a Convolutional Neural Network (CNN). Specifically, in our experiments, we employ MobileNetV2 as the backbone architecture. Local features are obtained by sampling the intermediate feature maps of MobileNetV2, capturing both low-level details and high-level semantic information. Each feature *x* is defined for a specific spatial location within a feature map and encapsulates the local context in that region. These vectors serve as the inputs to our cascaded anomaly detection pipeline, where they are evaluated independently at each stage to compute an anomaly score.

MobileNetV2 provides a hierarchy of feature maps that capture both localized texture and global semantic patterns, making it well-suited for anomaly detection in texture-rich and object category images. For a given layer ‘*l*’ of MobileNetV2, the feature map is denoted by(1)$$F^{\left(l\right)}\in R^{H_{l}\times W_{l}\times C_{l}},$$
where $H_{l}$ and $W_{l}$ are the height and width, and $C_{l}$ is the number of channels in that particular layer. The number of channels in each layer is fixed in the original MobileNetV2 design [26] for optimal performance on ImageNet classification. For anomaly detection task, these predefined and pretrained channel features are utilized without modifying the architecture. In order to detect and localize an anomaly, a local feature vector x is then defined by selecting the activations from a specific spatial location (i,j) within this feature map:(2)$$x=F^{\left(l\right)}\left(i,j\right)\in R^{C_{l}},\;i=1,\ldots ,H_{l},\;j=1,\ldots ,W_{l}.$$

Thus, x is the locally extracted feature vector of dimensions $C_{l}$ from one of the layers and serves as the primary input for our cascaded localized anomaly detection pipeline.

A full-image anomaly map is generated by collecting the anomaly score at each spatial location and subsequently upscaling to match the size of the input image. Figure 1 depicts the whole process starting from the input image, extraction of feature maps from different layers of the CNN, corresponding PCA transform for each feature vector, course pixel-level anomaly map generation and subsequently image-level classification through a cascaded detector mechanism. In the following three sub-sections, we describe the proposed PCA-based anomaly detector module, cascaded multi-stage overall system and the associated pre/post-processing steps respectively in more details.

### 3.1. Leveraging Near-Zero Variance Principal Components for Anomaly Detection

The main idea behind the anomaly detection suggested in PaDiM is to learn the *Normality* from the image features as a Multivariate Gaussian model. Thus, for all spatial feature vectors extracted from different layers of the particular CNN, a multivariate Gaussian distribution is assumed and modeled as:(3)$$p\left(x\right)=\frac{1}{\sqrt{{\left(2\pi \right)}^{d}\left|\Sigma \right|}}\exp\left(-\frac{1}{2}{\left(x-\mu \right)}^{T}\Sigma^{-1}\left(x-\mu \right)\right),$$
where μ and Σ are the mean vector and covariance matrix, respectively. ‘d’ is the number of dimensions of the feature vector x and is equal to $C_{l}$ i.e., the number of channels in that particular CNN layer from which this feature has been extracted.

From a training set of *n* normal images, these parameters are estimated as follows:
Mean:
(4)$$\hat{\mu }=\frac{1}{n}\sum_{i=1}^{n}x_{i},$$
Covariance Matrix:
(5)$$\hat{\Sigma }=\frac{1}{n-1}\sum_{i=1}^{n}\left(x_{i}-\hat{\mu }\right){\left(x_{i}-\hat{\mu }\right)}^{T}.$$

Since, these features may be correlated (non-zero off-diagonal elements in the covariance matrix $\hat{\Sigma }$), it is suggested to use *Mahalanobis Distance* to determine the level of anomaly in a test vector ‘y’, extracted from the test image, as follows:(6)$$D_{M}\left(y\right)=\sqrt{{\left(y-\hat{\mu }\right)}^{T}{\hat{\Sigma }}^{-1}\left(y-\hat{\mu }\right)}.$$

Since feature correlations are captured in $\hat{\Sigma }$, a higher value of Mahalanobis distance indicates that a test sample lies far from the distribution of normal data. A threshold τ can be set such that:(7)$$\left\{\begin{array}{c} D_{M}\left(y\right)\geq \tau \;\Rightarrow \;y\;\mathrm{is}\;\mathrm{considered}\;\mathrm{anomalous}\\ D_{M}\left(y\right)<\tau \;\Rightarrow \;y\;\mathrm{is}\;\mathrm{considered}\;\mathrm{anomaly}\text{-}\mathrm{free} \end{array}\right.$$

However, this method involves run-time determination of the inverse of covariance matrix $\hat{\Sigma }$ which can severely impact the computational performance of the overall anomaly detection system especially when the number of involved features is high. Thus, to circumvent this problem, we suggest finding the Principal Component Analysis (PCA) transform for both train and test features. Generally, PCA has been considered by multiple machine learning approaches to reduce the number of significant features based on their respective variances (eigen values). The reduction in number of features has been observed to not only reduce computational requirements but improve performance as well in detection tasks especially in the wake of fewer training examples [27]. For the anomaly detection problem at hand, however, researchers have not found utility in the reduction of number of features through PCA. Thus, PaDiM [16] reduces computational complexity through *randomly* selected features since PCA-based selection of significant features (higher corresponding variance/eigen values) led to lower detection performance. However, our approach for using PCA takes a different route in addressing the anomaly detection task.

As mentioned earlier, traditional applications of PCA emphasize the importance of *high-variance* components, assuming that dominant eigenvalues capture the most meaningful structure in data. However, our empirical findings suggest that it is the *low-variance* (near-zero) principal components that exhibit heightened sensitivity to anomalous patterns and actually play a critical role in distinguishing anomalous instances. Theoretically, anomalies often manifest as deviations from the statistical norms established by normal data distribution. Because PCA constructs an *orthogonal basis* that maximizes variance, low-variance principal components encode variations that are *suppressed in normal data but exaggerated in anomalies*. Our observations indicate that anomalous test samples project onto these components with *higher coefficients*, suggesting that anomalies are disproportionately represented in subspaces that normal data does not frequently occupy. This leads to accurate detection of anomaly if these low-variance features are employed instead of the high-variance ones.

In this regard, our proposed strategy is similar to that of autoencoders which enforce a compact latent bottleneck [28]. This approach hinges on deliberately restricting the model’s representational capacity so that normal instances are reconstructed accurately while anomalies—lying outside the learned subspace—exhibit large reconstruction errors. Thus, given a dataset of *d*-dimensional feature vectors $X={\left\{x_{i}\right\}}_{i=1}^{n}$, we compute the mean vector, $\hat{\mu }$, and the covariance matrix, $\hat{\Sigma }$. An eigen decomposition of the latter, $\hat{\Sigma }=W\Lambda W^{⊤}$, yields orthonormal eigenvectors $W=[w_{1},\ldots ,w_{d}]$ and corresponding eigenvalues $\Lambda =\mathrm{diag}(\lambda_{1},\ldots ,\lambda_{d})$, where $\lambda_{1}\geq \ldots \geq \lambda_{d}$.

Conventionally, it has been suggested to retain the top few high-variance components for dimensionality reduction while keeping the most energy for better reconstruction of the signal. Thus, retaining the top *k* components (eigen vectors) with the highest variances (eigen values), a transform can be defined as:(8)$$W_{\mathrm{high}}=\left[\;w_{1},w_{2},\ldots ,w_{k}\;\right],\;k\ll d.$$

Projecting *x* onto this high-variance subspace with the most energy and reconstructing gives(9)$${\hat{x}}_{\mathrm{PCA}}\;=\;W_{\mathrm{high}}\;W_{\mathrm{high}}^{⊤}\;\left(x-\hat{\mu }\right)+\hat{\mu }$$

Since, the transform has been obtained from the features of anomaly-free training images, this reconstructed signal is deemed to be closer or ideally identical to the original signal for the features from the test images as well if the assumption of these belonging to the same Gaussian distribution holds true. Thus, the reconstruction error defined below should be close to zero for anomaly-free test images.(10)$$S_{\mathrm{PCA}}\left(x\right)\;=\;{∥x-{\hat{x}}_{\mathrm{PCA}}∥}_{2}^{2}.$$

This reconstruction error, i.e., the $\ell_{2}$-distance in the original space, can be used as the anomaly score as well. This is because under this formulation, normal samples—whose variation is concentrated in the high-variance directions—yield small $S_{\mathrm{PCA}}$, whereas anomalies that activate suppressed, low-variance modes produce large errors.

In anomaly detection, the score distributions can vary across categories. Moreover, even the test normal images may yield non-zero scores due to domain shifts or subtle variations not present in training images. Thus, a fixed threshold to detect anomaly is not workable. Due to this reason, earlier works such as PaDiM [16] determine the score threshold to distinguish normal from anomalous samples using the test images only. Since empirically, true anomalous images tend to produce significantly higher scores compared to normal ones. Therefore, a decision threshold is selected based on relative score statistics computed across validation data to optimize detection performance and minimize false positives.

This PCA-based framework essentially creates a “bottleneck” principle very commonly used in autoencoders. Thus, PCA’s high-variance subspace and the autoencoder’s latent code each define a restricted representational subspace for normal data. However, it should be noted that the PCA approach is strictly linear, carving out a *k*-dimensional hyperplane in $R^{d}$, while the autoencoder can learn a nonlinear manifold of dimension *k*. In both cases, the reconstruction error serves as an effective anomaly score: normal data yield small S(x), anomalies—lying outside the learned subspace—produce large S(x).

In contrast to multivariate Gaussian methods (e.g., PaDiM) that require explicit inversion of the covariance matrix and computation of the Mahalanobis distance, the PCA–based approach avoids the expensive $\Sigma^{-1}$ step. Since k≪d, each score evaluation costs only O(dk) operations (matrix–vector multiplies) rather than the $O(d^{3})$ required to invert Σ. This reduction in both algorithmic complexity and numerical instability leads to substantial speed–ups in high–dimensional settings without sacrificing anomaly detection performance.

Moreover, our experiments reveal that when the feature dimensionality *d* is large (several hundred features extracted from multiple CNN layers), many of the *m* smallest eigenvalues $\left\{\lambda_{d-m+1},\ldots ,\lambda_{d}\right\}$ become numerically negligible (i.e., $\lambda_{j}\approx 0$). Consequently, the corresponding eigenvectors span an *approximate null space* of the normal data manifold. For any normal sample x, its projection onto this subspace,(11)$$z_{\mathrm{low}}\;=\;W_{\mathrm{low}}^{⊤}\;\left(x-\hat{\mu }\right),\;W_{\mathrm{low}}=\left[\;w_{d-m+1},\ldots ,w_{d}\;\right]\in R^{d\times m},$$
is nearly zero in every coordinate.

This observation allows us to bypass full reconstruction for anomaly detection and directly define an anomaly score as the $\ell_{2}$-distance of these low-variance (near-zero) projections:(12)$$S_{\mathrm{null}}\left(x\right)\;=\;{∥z_{\mathrm{low}}∥}_{2}^{2}\;=\;{∥W_{\mathrm{low}}^{⊤}\;\left(x-\hat{\mu }\right)∥}_{2}^{2}.$$

Under this scheme, normal instances yield $S_{\mathrm{null}}\left(x\right)\approx 0$, while anomalies—having nonzero components in the suppressed subspace—produce $S_{\mathrm{null}}\left(x\right)\gg 0$. This further reduces computation to a single matrix–vector multiply and norm calculation (O(dm)), eliminating the need for covariance inversion, reconstruction, or explicit error evaluation.

Our experimental results demonstrate that this approach of using components with near-zero variances not only dramatically reduces the computational complexity of the anomaly detector compared to Mahalanobis distance-based PaDiM, but also improves the detection accuracy as well. This improvement in accuracy may be attributed to the identification of suppressed null subspace using this proposed approach. Our ablation study has indicated that keeping upto 70% of the components (with corresponding $\lambda_{j}\approx 0$) leads to optimal anomaly detection performance which is also indicative of the size of the null space for the corresponding MobileNetV2 feature maps.

It may be noted that while this proposed anomaly detection approach avoids the computational overhead of matrix inversion inherent in Mahalanobis distance by leveraging eigenvalues derived from the PCA-based null space, the PCA decomposition is indeed required during training. This cost is, however, relatively minor compared to the intensive process of end-to-end CNN training found in models like SimpleNet and SuperSimpleNet as reported in the results section.

### 3.2. Cascaded Multi-Stage Anomaly Detection

Earlier works on CNN-based anomaly detection have often relied on combining features from different layers into a single, unified representation. In contrast, we propose a cascaded approach wherein anomaly detection from the features extracted from each CNN layer are considered as independent stages in a cascade. Each stage independently evaluates an anomaly score and applies a set of thresholds to filter samples.

Let $S_{s}\left(x\right)$ denote the anomaly score map computed at stage *s* constructed from the spatial anomaly scores. For each stage, two thresholds are defined: a lower threshold $T_{s}^{\mathrm{low}}$ and an upper threshold $T_{s}^{\mathrm{high}}$. The decision rule at stage *s* is as follows:(13)$$\left\{\begin{array}{cc} \mathrm{If}\;\max\left[S_{s}\left(x\right)\right]\leq T_{s}^{\mathrm{low}}, & \mathrm{classify}\;S_{s}\left(x\right)\;\mathrm{as}\;\mathrm{normal}\;\mathrm{and}\;\mathrm{discard}\;\mathrm{it},\\ \mathrm{If}\;\max\left[S_{s}\left(x\right)\right]\geq T_{s}^{\mathrm{high}}, & \mathrm{classify}\;S_{s}\left(x\right)\;\mathrm{as}\;\mathrm{anomalous}\;\mathrm{and}\;\mathrm{terminate}\;\mathrm{further}\;\mathrm{processing},\\ \mathrm{Otherwise}, & \mathrm{go}\;\mathrm{to}\;\mathrm{the}\;\mathrm{next}\;\mathrm{stage}\;\mathrm{for}\;\mathrm{further}\;\mathrm{analysis}. \end{array}\right.$$

This cascading mechanism ensures that samples with confident normal or anomalous scores at any stage are immediately classified, while only the ambiguous cases continue through the subsequent layers.

This framework offers two key advantages: First, it simplifies computations by restricting extensive processing only to those samples that are not clearly normal or anomalous at a given stage, thereby reducing the overall computational load. Second, by treating each layer independently, the cascade promotes localized decision-making. This leads to improved overall accuracy as the hierarchical structure allows for progressive refinement of the anomaly decision, leveraging the unique information captured at each CNN layer suppressing each other’s false positive and false negative rates. Moreover, in earlier works where feature maps from different layers are combined to make a single detector, upsampling of certain feature maps is required owing to different spatial size for different layers. This operation has been rendered redundant in our proposed approach. According to our ablation study, using four stages with feature maps from different layers of MobileNetV2 gives the optimal performance.

### 3.3. Image Pre- and Post-Processing

In our pre-processing pipeline, we first resize input images to 256×256 and then center-crop them to 224×224 to match the CNN’s input size, which is consistent with established techniques in the literature. However, we observed that in the MVTec dataset, several classes have original resolutions below 1024×1024. For example, the ‘tile’ class measures 840×840, ‘bottle’ 900×900, ‘metalnut’ 700×700, and ‘pill’ 800×800. Directly rescaling these smaller images leads to undesirable distortions. To address this issue, we add padding to such images to upscale them uniformly to 1024×1024 before performing the resizing and cropping operations. This additional padding step preserves the aspect ratio, reduces distortion, and ultimately contributes to higher overall accuracy. It may be noted that although reducing image resolution may sacrifice some information, this work adheres to a well-established anomaly detection paradigm where benchmark datasets provide sufficiently high-resolution images, ensuring that resizing doesn’t hinder detection. This is in line with the fact that the empirical evidence across anomaly and general object detection literature consistently supports using reduced image sizes with pretrained ImageNet-based CNN features over full-size images with custom extractors.

To mitigate the issue of limited training samples—which, for instance, results in rank-deficient data matrices for classes like ‘toothbrush’ (MVTec dataset) with only 60 images—we employ image augmentations. Specifically, we apply a 10% zoom-in along with random rotations of $\pm 2.5^{\circ }$. This augmentation strategy not only enriches the training set but also ensures that the null space is not overly stretched by minor variations; as a result, only the most significant anomalies result in notable deviations, while common alterations, such as slight zooming and displacements, are appropriately included within normal variations.

After obtaining the anomaly maps at each cascade stage, we perform post-processing to align the output with the input image dimensions. Specifically, the anomaly maps are upsampled to match the original image size, and a Gaussian smoothing filter with σ=4 is applied, as in the PaDiM approach. This smoothing operation yields more coherent anomaly boundaries.

## 4. Results

We evaluate our proposed anomaly detection framework on two widely used industrial benchmarks: the MVTec AD dataset and the VisA dataset. These datasets encompass a diverse range of object categories and texture patterns, making them suitable for assessing the generalizability and robustness of our approach. The reported results are in line with established anomaly detection literature where these datasets are considered since these are purpose-built for industrial quality control applications and are characterized by their controlled imaging environments, particularly in terms of consistent object positioning and alignment across samples. Since comparable reference works also evaluate their methods under these same assumptions, we restrict our analysis to these controlled settings to ensure fair and meaningful performance comparisons. The MVTec AD dataset is tailored for industrial visual inspection and contains 5354 high-resolution RGB images, typically sized at 1024 × 1024 pixels. It’s organized into 15 object categories, split between textures (e.g., carpet, grid, wood) and manufactured items (e.g., bottle, cable, pill, and transistor). Each category consists of defect-free training images and a test set containing both normal and anomalous samples. The training set has 3629 exclusively normal images. The test set includes a total of 1725 images, with 1528 anomalous and 467 normal ones. Anomalies range from tiny scratches (Capsule) and spots (Pill) to missing (Toothbrush) or damaged (Hazelnut) parts or deformations (Transistor), and vary in scale—from subtle surface variations (Wood) covering a few pixels to large structural flaws (Tile) occupying over 30% of the object’s area. Pixel-level ground truth annotations highlight anomaly regions, facilitating segmentation tasks. Images are captured under controlled lighting with industrial cameras to ensure consistent quality. On the other hand, the VisA dataset, is larger and offers more category complexity. It includes 10,821 RGB images across 12 object categories, typically sized around 1024 × 1024 or higher resolution, depending on the sub-dataset. These categories span multi-instance consumer goods (e.g., capsules, macaroni, chewing gum), single-item food products (cashew), and complex PCB assemblies. Image annotations include both image-level labels for anomaly classification and pixel-level masks for segmentation just like MVTec AD dataset. Anomalies are diverse, ranging from micro-defects like punctures (Chewinggum) and stains (Candle) to macro-anomalies like broken (Cashew), missing, or misplaced parts (PCB1, PCB2, PCB3, PCB4), with size variations from less than 1% to more than 40% of the object area. Training sets contain only normal images, with class-specific volumes ranging from a few hundred to over 1000 samples per category. The test sets include both normal and anomalous images, with roughly a 70:30 ratio, varying by each particular subset. For each dataset, we report class-wise and aggregate metrics i.e., area under the ROC curve (AUROC) on per-image and per-pixel basis, to facilitate comparison with prior methods. Additionally, a runtime analysis has been provided to highlight the computational advantages of using MobileNetV2 as backbone for anomaly detection combined with the lightweight PCA-based cascaded detectors. We also present ablation studies to assess the individual contributions of our key components i.e., the percentage sized of the PCA null subspace, the number and specific CNN layers for the cascaded architecture and the effect of image augmentations. These experiments demonstrate that our approach not only achieves state-of-the-art performance using a compact network but also provides noticeable improvements in inference speed and efficiency, making it well-suited for real-time industrial inspection scenarios.

### 4.1. Experimental Setup

All experiments reported in this section were conducted in MATLAB R2023 environment on a Windows machine equipped with an AMD Ryzen 5 5600H CPU @ 3.5 GHz, 8 GB of RAM, and an NVIDIA GeForce RTX 3050 GPU. Our evaluation focuses on two primary metrics: per-pixel AUROC and per-image AUROC, which are widely adopted in existing literature [15,16]. Although some recent methods report additional metrics such as the PRO-score (per-region overlap), we align our evaluation protocol with prominent baselines such as PaDiM, PatchCore, and SimpleNet, which rely solely on AUROC-based measures. This consistency ensures fair and direct comparison with state-of-the-art approaches.

### 4.2. Anomaly Detection on Standard Datasets

Table 1 presents the anomaly detection performance of our proposed method on the MVTec AD dataset across 15 object and texture categories in terms of both image-wise and pixel-wise AUROC. For each category, three different experiments were run to ensure different augmentations of all the images in this category were used to train the PCA null-space. Notably, our proposed approach achieves a perfect detection rate (100% per-image AUROC) on 7 of the 15 categories, including *carpet*, *leather*, *wood*, *bottle*, *hazelnut*, *toothbrush*, and *transistor*, demonstrating its strong capability in both texture and object-centric settings. For the remaining categories, the method maintains consistently high accuracy, with minimum AUROC value as 96.7% (Pill) which is still better than PaDiM, FastFlow [4,29] and PatchCore detectors.

Although the overall image-level AUROC is marginally lower than recent approaches such as SimpleNet and GLASS, the difference is limited to only 0.5%. Given the relatively small size of the MVTec dataset (*1725* test images), such a narrow margin is within the range of natural statistical variance and does not significantly impact the reliability or robustness of our method. These results affirm the effectiveness of our lightweight, cascaded PCA-based detection pipeline in achieving near state-of-the-art detection performance while maintaining considerable computational efficiency.

It may be observed that the consistently low standard deviations across the three experiments for both image-level and pixel-level AUROC values in each category indicate a high level of statistical reliability and significance.

As shown in Table 2, on the VisA dataset, our proposed method demonstrates competitive performance across all object categories, achieving state-of-the-art results on *pcb4*. While our overall AUROC score is slightly behind SuperSimpleNet by only 1.7%, this difference is relatively insignificant considering the limited number of anomalous samples in the dataset—only 1200 images—making such marginal gaps subject to dataset variance. Notably, our method still outperforms SimpleNet, underscoring its robustness and efficiency despite the use of a simpler backbone i.e., MobileNetV2 network compared to ResNet50 for all the other reference works i.e., PaDiM, PatchCore, SimpleNet and SuperSimpleNet. The results pertaining to PaDiM (based on Gaussian modeling and Mahalanobis distance) on VisA dataset have been generated by ourselves since these were not reported by the original authors. All other results have been taken directly from the published sources.

Although PatchCore achieves the highest overall accuracy on this dataset, it does so at the cost of significant computational complexity and slow inference times, as will be discussed in the subsequent runtime analysis. By contrast, our approach strikes a desirable balance between speed and accuracy, making it better suited for real-world deployment where latency and resource constraints are critical factors.

It is important to note that per-pixel AUROC results for the VisA dataset are not available for the considered reference methods in the literature, making a fair, comprehensive comparison at the pixel level infeasible. Similarly, for the MVTec dataset, SuperSimpleNet does not report per-pixel AUROC scores, and as such, we omit these values in our evaluation. To maintain consistency and ensure meaningful comparisons, we primarily focus on the per-image AUROC metric, which is widely reported and reliably benchmarked across all evaluated methods.

Figure 2 and Figure 3 provide a visual overview of our anomaly detection approach applied to sample images from the MVTec AD and VisA datasets. For each class, the input image is shown alongside its corresponding ground truth anomaly mask and the resulting anomaly heatmap generated by our method. These qualitative results demonstrate that our approach reliably localizes and delineates anomalous regions across diverse settings. While the heatmaps generally align well with ground truth masks, they are not pixel-perfect due to the remapping of low-resolution CNN feature maps to the original image scale. To objectively capture this spatial imprecision, Table 1 has reported pixel-level (AUROC) scores as well, providing a quantitative measure of anomaly localization quality. Despite this inherent limitation, the detector remains highly effective in practical industrial applications where precise pixel-wise accuracy is not a strict requirement.

### 4.3. Inference Speed Comparison

The speed analysis summarized in Table 3 highlights the computational efficiency of the proposed anomaly detection framework. Achieving a processing speed of 20.1 frames per second (fps) on an NVIDIA RTX 3050 GPU, our method surpasses all referenced techniques, including SimpleNet (17.3 fps), PatchCore with ResNet18 (11.1 fps) and ResNet50 (9.0 fps), and PaDiM (3.8 fps). This substantial speed advantage is primarily due to the use of MobileNetV2, a lightweight architecture with significantly fewer parameters and lower memory requirements compared to deeper networks like ResNet50. While methods such as PatchCore offer strong detection performance, their inference times are notably slower, making them less suitable for latency-sensitive applications. In contrast, the proposed cascaded low-variance PCA detector not only achieves comparable or superior accuracy but also delivers the fastest inference among all evaluated approaches. This balance between speed and effectiveness makes it particularly suitable for real-time deployment in industrial inspection scenarios where both responsiveness and computational efficiency are critical.

In summary, our proposed approach demonstrates impressive real-time performance by achieving around 20 frames per second on an NVIDIA RTX 3050 GPU—a widely accessible, mid-range hardware platform. While other methods reported in the literature show higher fps values, they usually rely on very high-end GPUs such as the NVIDIA Tesla V100S (SuperSimpleNet) or RTX 3080 Ti (SimpleNet), which are not as readily available or cost-effective for industrial applications. The efficiency of our method stems from the lightweight MobileNetV2 backbone and the cascaded PCA-based detection pipeline, which together minimize computational overhead without sacrificing detection accuracy.

It should be noted that the reported processing speed of 20.1 fps for our proposed detector corresponds to the worst-case scenario where every cascade stage is executed. In practical deployment, many samples are filtered out in the early stages of the cascade, meaning that the effective inference speed for the majority of images is actually higher. By reporting the full cascade execution times, we emphasize the robustness of our method even for the most challenging cases, while also underscoring its potential for even greater efficiency in real-world scenarios.

Moreover, empirical observations show that all 15 categories of the MVTec AD dataset—comprising 3629 training images—can be fully processed within 30 min. Notably, it has been observed that the PCA decomposition task takes a quarter of time than the associated feature extraction from CNN layers. This highlights the computational efficiency of the proposed method, even during training, reinforcing its practicality for real-world deployment.

### 4.4. Ablation Study

This section presents ablation study experiments used to select various parameters of the proposed detector.

#### 4.4.1. CNN Backbone Selection

Table 4 depicts experimental results of different CNN architectures considered in this study. This selection of lightweight CNN backbones were tested on the MVTec AD dataset, assessing their anomaly detection performance across three critical dimensions: AUROC, processing speed (fps), and model parameter count. Among the candidates, MobileNetV2-Block4 (features extracted from Block4 layer) emerged as the most favorable architecture. It achieved the highest image-level AUROC score, demonstrating robust detection capabilities, while also maintaining high inference speed (20.1 fps) and an exceptionally compact memory footprint (42 K parameters). This combination positions MobileNetV2-Block4 as a well-balanced model for real-time industrial applications, offering precision and efficiency without compromising deployability on resource-constrained platforms. Its architectural simplicity and early layer truncation appear to contribute positively to both speed and representation quality, making it an ideal backbone for lightweight anomaly detection frameworks. While SqueezeNet turns out to be the fastest, it’s detection performance is sub-optimal. Similarly, ResNet18-Block3 and ResNet18-Block4 fall short in detection accuracy. Since, our ultimate detector utilizes multiple layers to create a cascade, we select MobileNetV2-Block13 which encompasses earlier layers i.e., Block11, Block 8 and Block4 as well. This creates a detector which leads to even higher AUROC than a single layer can achieve.

#### 4.4.2. Role of Specific CNN Layers

Table 5 details the per-class AUROC values for individual MobileNetV2 layers when used as independent detectors on the MVTec dataset. Notably, even the early-stage Block 4 achieves a respectable AUROC of 96.2%, demonstrating its ability to capture local features important for anomaly detection. Block 8, on the other hand, markedly outperforms the others with an AUROC of 98.5%, indicating its strong discriminative power. In contrast, deeper layers such as Block 11 and Block 12 yield 94% and 96.1% respectively, while Block 13 falls back to 94%. These observations reveal that while Block 8 alone could deliver the best results, incorporating features from both early (Block 4) and deep (Block 13) stages provides complementary local and global perspectives. Therefore, our proposed cascade design, depicted in Figure 1 strategically integrates feature maps from Block 4, Block 8, Block 11 and Block 13 layers of MobileNetV2 CNN to exploit the strengths of each stage, ensuring robust and balanced anomaly detection.

#### 4.4.3. PCA Null-Space Size Across Different CNN Layers

Figure 4 presents the eigenvalue distributions for three distinct classes—screw, carpet, and leather—across layers Block 11, Block 8, Block 7, and Block 3. In each plot, the eigenvalues computed from the PCA of the feature maps are shown simultaneously, revealing that a significant proportion of the eigenvalues are very close to zero. In fact, nearly 70% of the eigenvalues across these layers exhibit minimal variation, indicating that these features carry little information related to normal images. This observation has been further substantiated by our experiments, which demonstrate that constructing the null-space based detector using the subset of features corresponding to the lowest 70% of eigenvalues yields optimal detection performance. Thus, our analysis validates that leveraging a reduced representation—focused on the low-variance components—effectively captures the essential characteristics required to distinguish anomalies from normal patterns. Table 6 shows the results of ablation study to select the PCA Null-Space size based on AUROC values obtained on MVTec AD dataset. It can be observed that the AUROC values peak when 70% of features corresponding to the smallest eigen values are selected and sharply declines on either side of this number. This is consistent with the observations made in Figure 4 and Figure 5.

Figure 5 depicts the detection scenario for the ‘Transistor’ and ‘Carpet’ categories in MVTec AD dataset using the eigen values data corresponding to the block-11-add layer for both anomalous and normal images. It can be seen that the eigen values corresponding to near-zero values (Null-Space) in the training data activate more for the anomalous images in the test data than for the normal images. The normal images in the test set, however, do depict non-zero values since these images were not part of the training set and exhibit subtle differences from the training images leading to deviation from null-space. This example effectively demonstrates how the proposed detector achieves good detection performance by focusing on the features which exhibit near-zero variance in the train data. The anomalies are effectively detected since these same features yield non-zero values in the test set.

#### 4.4.4. Impact of Image Augmentation

Incorporating image augmentation while computing the PCA transformation has proven highly beneficial as per our experimentation. Without augmentation, we observed an overall drop of up to 0.6% in the average AUROC value. This impact is particularly pronounced for classes like *toothbrush*, which presents only 60 training images, leading to a rank-deficient PCA covariance matrix and consequently less accurate anomaly detection. To address this, we applied simple augmentation techniques—specifically, a 10% zoom-in coupled with random rotations of $\pm 2.5^{\circ }$—to generate augmented normal images with sufficient variability. Our experiments demonstrated that these augmentations substantially improved performance in challenging classes, including *grid, cable, pill, capsule,* and *screw*, thereby reinforcing the critical role of data diversity in achieving robust anomaly detection.

### 4.5. Threshold Selection in the Cascaded Architecture

As mentioned earlier, in anomaly detection task, score distributions usually differ across different classes even in the same dataset due to their very different natural shapes. Additionally, even test normal images can produce non-zero scores due to domain shifts or minor variations absent in training data, making a fixed threshold for anomaly detection impractical. Consequently, methods like PaDiM [16] use test images to set the score threshold for distinguishing normal from anomalous samples. Since true anomalous images typically generate significantly higher scores than normal ones, a decision threshold is chosen based on relative score statistics from validation data to enhance detection accuracy and reduce false positives. In the proposed cascaded detector, however, each stage needs to select two thresholds for operation as discussed earlier.

Figure 6 illustrates the process of selecting the upper and lower thresholds for a given stage in our cascaded detector design (Equation (Equation 13)) using anomaly scores obtained from the test data. The lower threshold is deliberately set to filter out examples that are clearly normal, thus reducing unnecessary processing in subsequent stages. Conversely, the upper threshold is chosen to directly identify and classify samples that exhibit clear anomalous characteristics. The examples that fall in between these two thresholds are deemed ambiguous, as the current detector cannot conclusively label them as either normal or anomalous; these cases are consequently passed on to the next stage of the cascade for further analysis. The anomaly scores used for this thresholding are extracted by taking the maximum value from the anomaly score maps produced by the PCA detector on the corresponding CNN feature map. This approach of determining the anomaly score is consistent with that of PaDiM and SimpleNet. This thresholding strategy allows each stage in the cascade to focus on processing only the most uncertain cases, ultimately enhancing the overall accuracy and efficiency of the anomaly detection system.

After every cascade stage, the number of ambiguous examples decreases significantly. In the initial stages, many samples fall into the ambiguous range between the clearly normal and clearly anomalous thresholds. However, as the cascade progresses, each stage accurately filters out a portion of these borderline cases. Ultimately, this cascading approach not only improves overall efficiency but also contributes to a higher precision in anomaly detection by dedicating more refined analysis to the few remaining ambiguous cases.

### 4.6. False Positive Analysis

Figure 7 and Figure 8 illustrate several false positive instances observed on the MVTec AD and VisA datasets. In many of these cases, our model erroneously labels certain background elements as anomalous. Many of these false positives arise from random objects or textures in the background that were not encountered during the training phase. This suggests that, although the model is highly effective at detecting true anomalies, it can also be sensitive to background variations that fall outside the realm of the training data. Such misinterpretations highlight the challenge of developing robust anomaly detection systems when the diversity of background scenarios is limited. Addressing this issue may involve incorporating more comprehensive background data during training or refining the model’s ability to distinguish between relevant defects and irrelevant environmental variations.

## 5. Conclusions and Future Directions

In conclusion, our work demonstrates that a lightweight approach utilizing MobileNetV2 coupled with a cascaded low-variance PCA detector can achieve competitive anomaly detection performance on challenging industrial datasets such as MVTec AD and VisA. Our method not only offers impressive real-time processing speeds but also maintains robust detection and localization of anomalies, making it highly practical for resource-constrained industrial applications. However, despite these strengths, our framework does exhibit certain limitations. Unlike methods such as SimpleNet and SuperSimpleNet, which incorporate additional training steps through feature adaptation and a dedicated anomaly discriminator that are fine-tuned on domain-specific data, our approach relies solely on fixed, pretrained features. This design choice, while making our approach easier to train, may restrict its adaptability to complex or evolving anomaly patterns that could benefit from learnable refinements.

Thus, future research could focus on integrating a hybrid training paradigm that combines the efficiency of pretrained MobileNetV2 with targeted fine-tuning in the later stages of the detection pipeline. Such an approach could help bridge the performance gap with heavily trained frameworks like SimpleNet and SuperSimpleNet, potentially leading to further improvements in both accuracy and robustness.

Moreover, integrating artificial anomaly generation techniques during training is a promising direction for further enhancing network robustness. Inspired by the synthetic anomaly generation approach of SuperSimpleNet, we propose exploring the use of controlled perturbations, such as the application of Perlin noise. Perlin noise, with its smooth yet naturally varied texture characteristics, can simulate subtle imperfections and localized irregularities that mimic real-world defects. By injecting Perlin noise into the training images, our framework could be exposed to a wider range of abnormal patterns, which in turn would encourage the model to learn more discriminative and generalized anomaly representations. This method not only alleviates the challenges posed by limited anomaly data but also offers an efficient strategy to enrich training datasets without extensive manual annotation.

Future work could also investigate the intrinsic visual characteristics or dataset-specific patterns that contributed to perfect anomaly detection performance in select categories, offering insights into model sensitivity and potential biases in feature representation.

Further work could also evaluate the model’s robustness in less controlled environments by leveraging a modified variant of the MVTec AD dataset, referred to as “Rd-MVTec AD.” This extended version introduces random rotations and cropping operations to emulate scenarios where objects are not consistently centered or aligned—conditions that reflect more realistic industrial settings beyond the constraints of traditional benchmarks considered in the present work.

## Figures and Tables

**Figure 1 sensors-25-04853-f001:**
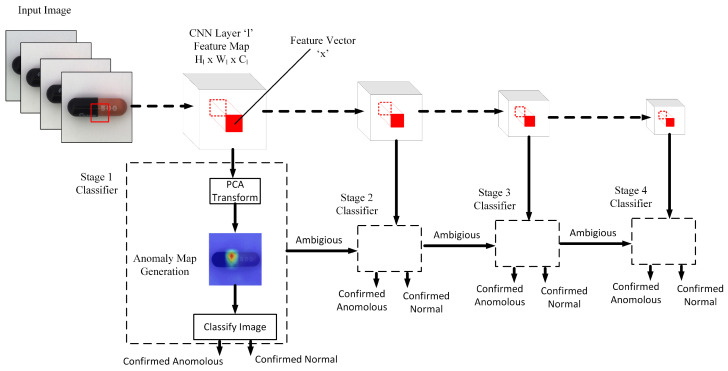
Proposed anomaly detector system: Spatial features are extracted from the feature maps of different layers and used by the cascade detectors to make anomaly/normal/ambiguous decision.

**Figure 2 sensors-25-04853-f002:**
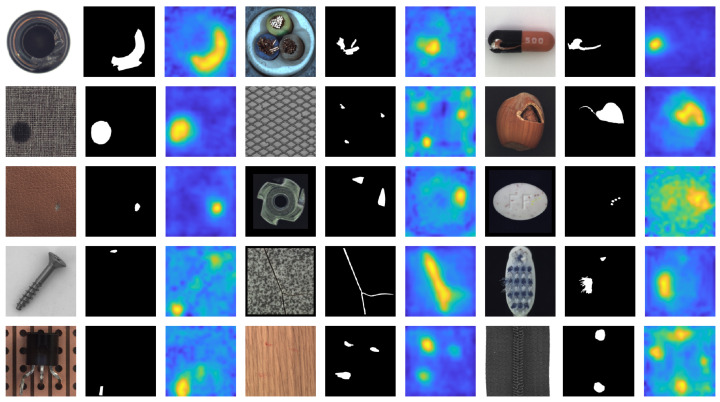
Visual results of anomaly detection on MVTec dataset using the proposed cascaded detector (input image, ground truth mask and the predictions heatmap).

**Figure 3 sensors-25-04853-f003:**
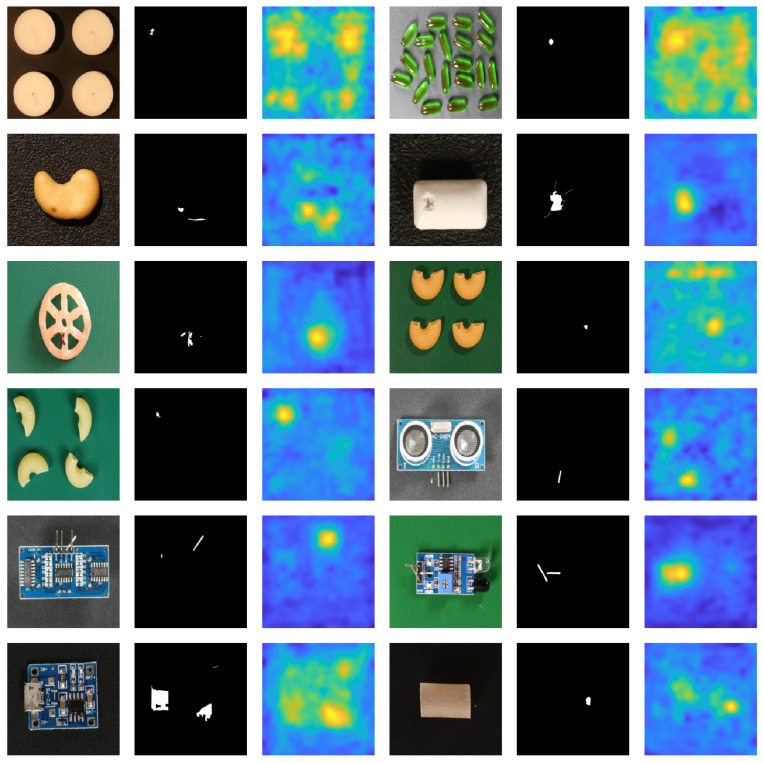
Visual results of anomaly detection on VisA dataset using the proposed cascaded detector (input image, ground truth mask and the predictions heatmap).

**Figure 4 sensors-25-04853-f004:**
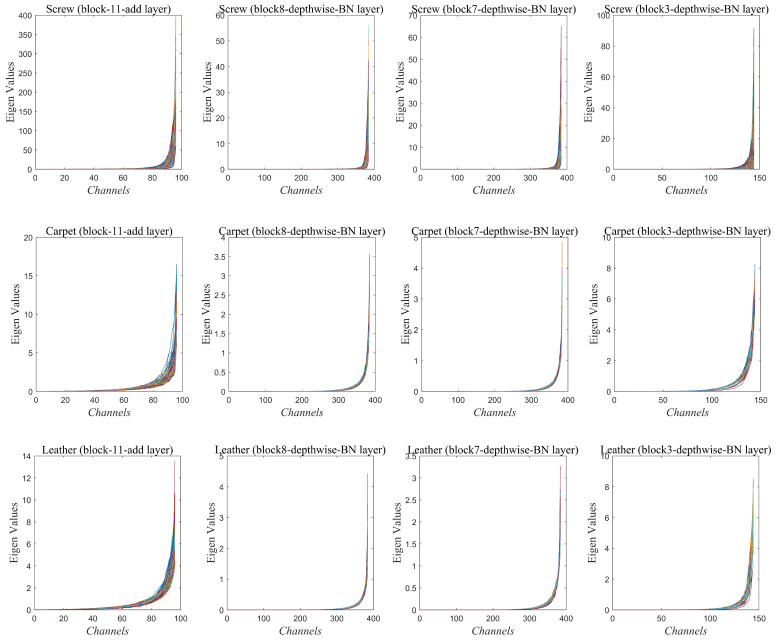
Demonstration of PCA null space: Features extracted from different layers of MobileNetV2 CNN for different classes of train data in MVTecAD dataset depict multiple near-zero eigen values. Each example is depicted using a different color.

**Figure 5 sensors-25-04853-f005:**
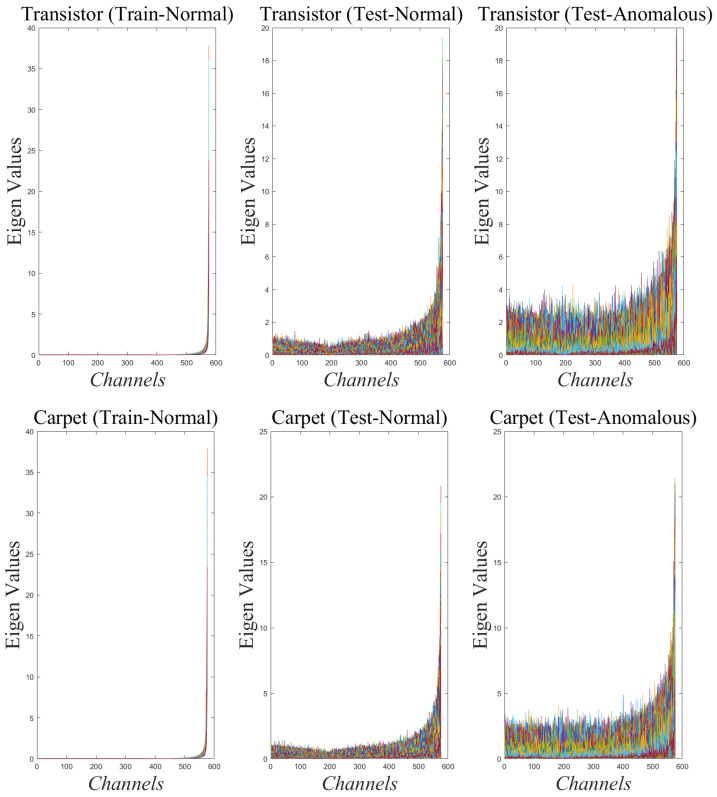
Demonstration of near-zero variance eigen values (MobileNetV2:block-11-add layer) for training data depicting different levels of activations for anomolous and normal test data for the Transistor and Carpet categories in MVTecAD dataset. Each example is depicted using a different color.

**Figure 6 sensors-25-04853-f006:**
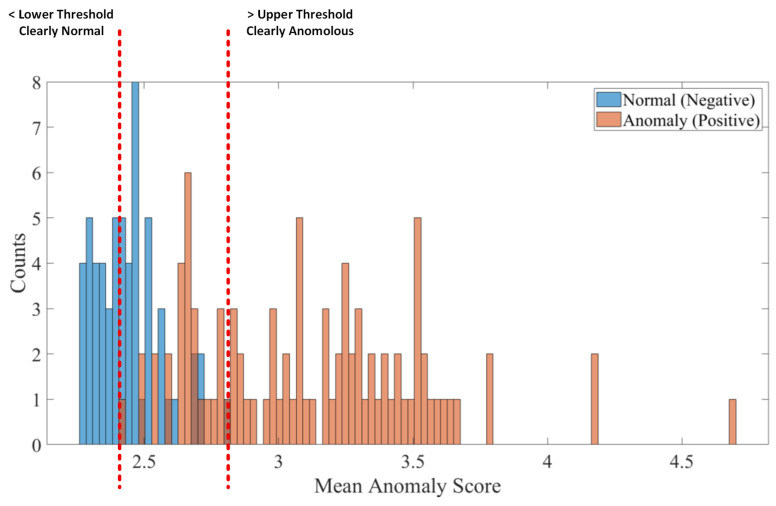
Determination of lower and upper threshold for a particular stage in the proposed cascaded detector.

**Figure 7 sensors-25-04853-f007:**
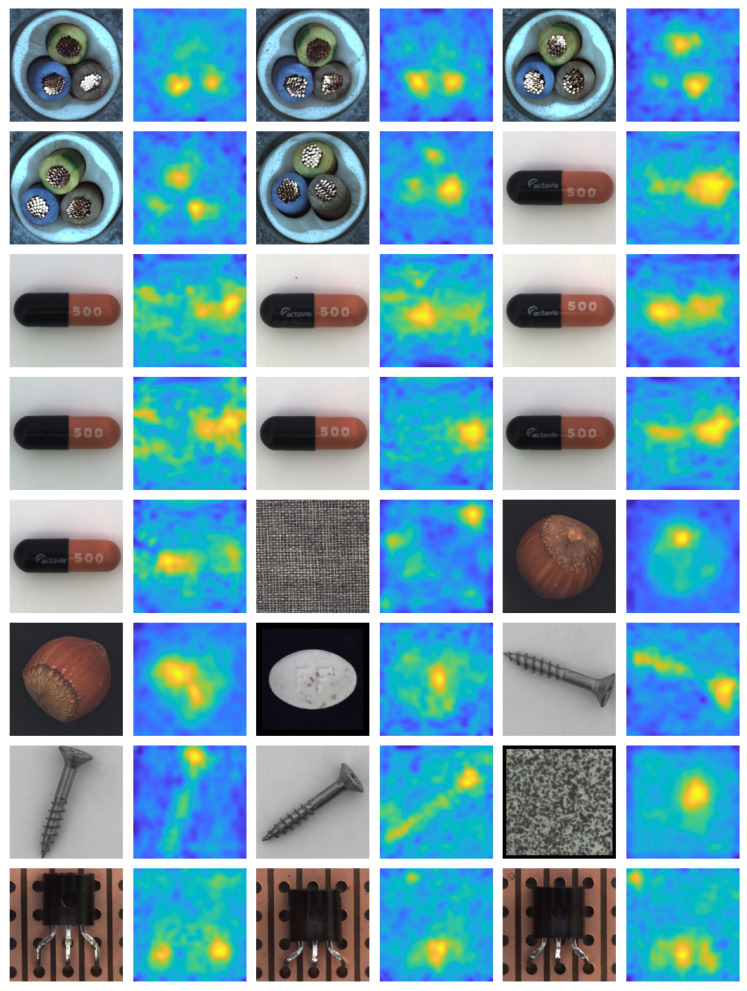
Sample false positive results for the proposed detector on MVTec dataset.

**Figure 8 sensors-25-04853-f008:**
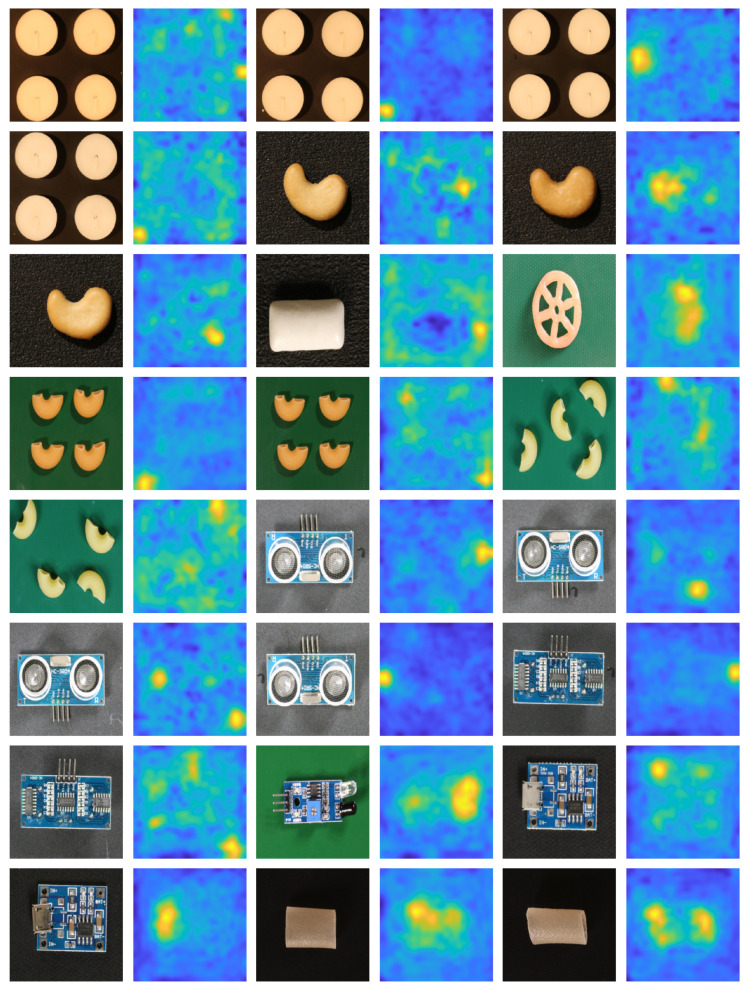
Sample false positive results for the proposed detector on VisA dataset.

**Table 1 sensors-25-04853-t001:** Comparison with the reference works on MVTec AD dataset for anomaly detection and localization (Image-wise AUROC/Pixel-wise AUROC).

Class	PaDiM	FastFlow	PatchCore	SimpleNet	SuperSimpleNet	GLASS	Proposed
Carpet	99.8/99.1	97.5/92.9	98.7/99.0	99.7/98.2	98.4/-	99.8/**99.6**	**100 ± 0.00**/98.4 ± 0.01
Grid	96.7/97.3	**100**/96.0	98.2/98.7	99.7/98.8	99.3/-	**100**/**99.4**	99.2 ± 0.13/97.2 ± 0.06
Leather	**100**/99.2	**100**/99.1	**100**/99.3	**100**/99.2	**100**/-	**100**/**99.8**	**100 ± 0.00**/98.6 ± 0.00
Tile	98.1/94.1	99.9/87.3	98.7/95.6	99.8/97.0	99.7/-	**100**/**99.7**	99.7 ± 0.11/93 ± 0.01
Wood	99.2/94.9	98.9/93.1	99.2/95.0	**100**/94.5	99.3/-	99.9/**98.8**	**100 ± 0.00**/93.6 ± 0.01
Bottle	99.1/98.3	**100**/89.3	**100**/98.6	**100**/98.0	**100**/-	**100**/**99.3**	**100 ± 0.00**/97.7 ± 0.00
Cable	97.1/96.7	93.9/89.9	99.5/98.4	**99.9**/97.6	98.1/-	99.8/**98.7**	99.1 ± 0.06/97.1 ± 0.00
Capsule	87.5/98.5	98.1/95.4	98.1/98.8	97.7/98.9	98.7/-	**99.9**/**99.4**	98.7 ± 1.28/98.4 ± 0.00
Hazelnut	99.4/98.2	98.9/95.6	**100**/98.7	**100**/97.9	99.8/-	**100**/**99.4**	**100 ± 0.00**/98.1 ± 0.00
Metal_nut	96.2/97.2	99.6/92.3	**100**/98.4	**100**/98.8	99.5/-	**100**/**99.4**	99.2 ± 0.05/98.5 ± 0.00
Pill	90.1/95.7	96.7/93.9	96.6/97.4	99.0/98.6	98.1/-	**99.3**/**99.4**	96.7 ± 0.21/98.4 ± 0.00
Screw	97.5/98.5	84.5/89.7	98.1/99.4	98.2/99.3	92.9/-	**100**/**99.5**	99.6 ± 0.37/98.7 ± 0.02
Toothbrush	**100**/98.8	89.2/87.0	**100**/98.7	99.7/98.5	92.2/-	**100**/**99.3**	**100 ± 0.00**/98.7 ± 0.00
Transistor	94.4/97.5	98.5/92.0	**100**/96.3	**100**/97.6	99.9/-	99.9/97.6	**100 ± 0.00**/**98.2 ± 0.02**
Zipper	98.6/98.5	98.5/93.7	99.4/98.8	99.9/98.9	99.6/-	**100**/**99.6**	98.7 ± 0.15/98.1 ± 0.00
Average	95.8/97.5	96.9/92.5	99.1/98.1	99.6/98.1	98.4/-	**99.9**/**99.3**	99.4 ± 0.16/97.5 ± 0.01

Note: Bold values depict the best results for each category.

**Table 2 sensors-25-04853-t002:** Comparison with the reference works on VisA dataset for anomaly detection task (Image-wise AUROC).

Class	PaDiM	FastFlow	PatchCore	SimpleNet	SuperSimpleNet	Proposed
Candle	95.9	96.8	**98.6**	92.5	97.1	97.5 ± 0.05
Capsules	64.2	**83.0**	76.4	78.9	81.5	70.2 ± 0.52
Cashew	89.9	90.0	**97.9**	91.9	93	94.1 ± 0.06
Chewing gum	99.5	**99.8**	98.	99	99.3	99.5 ± 0.00
Fryum	88.1	**98.6**	94.8	95.4	96.8	93.9 ± 0.02
Macaroni 1	81.6	94.8	**95.8**	94.2	93.1	88.1 ± 0.05
Macaroni 2	70.3	**80.5**	77.7	71.8	75	77.0 ± 1.02
PCB 1	95.7	95.5	**98.9**	92.5	96.9	96.1 ± 0.03
PCB 2	89.3	96.1	97.1	93.6	**97.5**	94.3 ± 0.02
PCB 3	79.3	94.0	**96.3**	92.6	94.4	90.5 ± 0.04
PCB 4	98.7	98.4	99.4	97.9	98.4	**99.7 ± 0.01**
Pipe fryum	95.9	99.6	**99.7**	94.6	97.6	99.6 ± 0.01
Average	87.4	93.9	**94.3**	91.2	93.4	91.7 ± 0.15

Note: Bold values depict the best results for each category.

**Table 3 sensors-25-04853-t003:** Speed comparison with the reference works on NVIDIA RTX 3050 GPU.

Detector	Backbone	Processing Speed (fps)
PaDiM	ResNet18	3.8
PatchCore	ResNet18	11.1
PatchCore	ResNet50	9.0
SimpleNet	ResNet50	17.3
Proposed	MobileNetV2	20.1

**Table 4 sensors-25-04853-t004:** Comparative analysis of lightweight CNN backbones for anomaly detection on MVTec AD dataset.

Backbone	AUROC	Processing Speed (fps)	Model Parameters
MobileNetV2-Block13	94.0	20.1	615 K
MobileNetV2-Block11	94.0	23.2	431 K
MobileNetV2-Block4	96.2	23.5	42 K
ResNet18-Block4	91.0	22.9	2.7 M
ResNet18-Block3	90.2	23.1	684 K
SqueezeNet-fire8	91.0	24.3	525 K

**Table 5 sensors-25-04853-t005:** Classification strength of different MobileNetV2 layers in terms of image-level AUROC on MVTec AD dataset.

Class	Block 4	Block 8	Block 11	Block 12	Block 13
	**192 Channels**	**384 Channels**	**576 Channels**	**576 Channels**	**576 Channels**
Carpet	97	98.2	100	99.5	98.5
Grid	95.1	98.5	97.2	95.2	70.8
Leather	99.9	100	100	100	100
Tile	98.9	99.2	98.7	97.6	98.3
Wood	99.6	99.5	98.8	99.3	99.2
Bottle	100	100	99.8	100	100
Cable	94.9	96.8	91.5	96.6	98.8
Capsule	90.5	96.1	84.4	91.2	93.3
Hazelnut	99.3	100	92	99.9	99.6
Metal_nut	96.1	98.5	96.6	97.6	96.6
Pill	95.2	93.9	81.3	88.8	88.5
Screw	86.5	98.2	79.8	82.4	77.2
Toothbrush	100	100	98.6	98.9	98.1
Transistor	98.5	100	99	100	99.3
Zipper	91.4	98.2	92.6	95	91.3
Average	96.2	98.5	94	96.1	94

**Table 6 sensors-25-04853-t006:** Selection of PCA null subspace size.

	100%	90%	75%	70%	65%
AUROC	95.4	97.2	99.1	99.4	98.9

## Data Availability

MVTec Anomaly Detection dataset can be downloaded from the URL https://www.mvtec.com/company/research/datasets/mvtec-ad (accessed on 22 June 2025). VisA Anomaly Dataset can be downloaded from the URL https://github.com/amazon-science/spot-diff (accessed on 22 June 2025). The source code for the proposed cascaded anomaly detector can be found at the URL https://github.com/4mbilal/Anomaly_Detection (accessed on 22 June 2025).

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
