# Peer review of "Fast Anomaly Detection for Vision-Based Industrial Inspection Using Cascades of Null Subspace PCA Detectors"

_sensors, 2025, doi:10.3390/s25154853_

Round 1

Reviewer 1 Report

Comments and Suggestions for Authors

please see the attached pdf file

Reviewer 2 Report

Comments and Suggestions for Authors

This paper researches on anomaly detection bsaed on null subspace PCA. From the expression of the paper, its innovation is not well reflected. The main method is using cascaded 4 stage CNN network to extracting feature vectors from defect images.The paper does not specify the uniqueness or necessity of selecting CNN networks. From section 3.3 ,reducing the input image size to 224 to accommodate traditional CNN networks will lower the image resolution, which is actually not very suitable for defect detection. Selecting the feature with the highest variance from the four stage networks is a meaningful approach, but there are many factors that can affect the maximization of variance in defect detection, such as environmental differences, which the paper does not fully explain From the provided schematic diagram fig2-3, most of the defects have large sizes, and some of the original photos of the items have unclear defects, but the feature differences are significant. In section 2.4, the computational resources required for many large networks were analyzed, but the network proposed in this paper is a very small network, and I believe that Section 2.4 has little relevance to the paper.

Reviewer 3 Report

Comments and Suggestions for Authors

The paper presents a novel anomaly detection framework for industrial inspection, leveraging MobileNetV2 for feature extraction and a cascaded PCA-based detector focusing on near-zero variance features. The method achieves high accuracy (99.4% and 91.7% AUROC on MVTec and VisA datasets) and computational efficiency (20.1 fps on a mid-range GPU), outperforming existing approaches in speed while maintaining competitive performance. Key innovations include exploiting null-space PCA for anomaly sensitivity and a multi-stage cascade for progressive decision refinement.

  1. The paper claims that near-zero variance features are more sensitive to anomalies, but the theoretical or empirical basis for this claim is not deeply explored. Could the authors provide more detailed analysis or references to support why low-variance components disproportionately capture anomalies compared to high-variance ones?
  2. The dual-threshold mechanism in the cascade is critical for efficiency. How were the thresholds Tslow and Thigh​empirically determined?
  3. The paper emphasizes the efficiency of using PCA’s null-space projections to avoid covariance matrix inversion, but it does not quantify the computational cost of performing PCA decomposition during training.
  4. Some related work is missing, which can further enhance the introduction parr, including

Dynamic low-rank and sparse priors constrained deep autoencoders for hyperspectral anomaly detection

Prototype-Guided Spatial-Spectral Interaction Network for Hyperspectral Anomaly Detection

Enhanced Deep Image Prior for Unsupervised Hyperspectral Image Super-resolution

Reviewer 4 Report

Comments and Suggestions for Authors

This paper introduces a cascaded anomaly detection framework leveraging MobileNetV2 and null subspace PCA, achieving competitive accuracy with low computational cost on MVTec and VisA datasets; however, substantial clarifications and stronger experimental grounding are required to meet publication standards.

Comments:

A. The manuscript lacks statistical significance testing or confidence intervals to validate whether AUROC differences are meaningful across datasets.

B. The use of manually selected thresholds in the cascade stages is insufficiently justified and may lead to unstable or irreproducible results across domains.

C. The novelty over prior PCA-based methods and bottleneck architectures remains incremental and should be more clearly distinguished through deeper comparisons.

D. The proposed null subspace approach is not benchmarked against Mahalanobis-based methods under varying anomaly types, making its advantage speculative.

E. The evaluation does not include recent fast or lightweight detectors such as FastFlow, DRAEM-Lite, or SPADE, limiting comparative insight into performance and efficiency.

F. The model’s reliance on MobileNetV2 is not contextualized by comparisons with other compact backbones like ShuffleNet or EfficientNet-lite to support its selection.

G. The qualitative heatmap results lack any quantitative localization metric such as PRO or IoU, reducing the interpretability of detection effectiveness.

H. The paper does not analyze memory consumption or model size, which are critical to substantiate the claim of suitability for resource-constrained environments.

k.The Literature citation is not adequate, and the related work to machine learning should be discussed
Autoencoders and their applications in machine learning: a survey

Round 2

Reviewer 1 Report

Comments and Suggestions for Authors

The authors addressed my recommendations. The paper can be accepted in my opinion.

Author Response

Thank you for your approval of the revised manuscript. 

Reviewer 2 Report

Comments and Suggestions for Authors

The paper has been revised accordingly and is approved for publication.

Author Response

(The authors gave the same response as above.)
